# Endometrial Injury Upregulates Expression of Receptivity Genes in Women with Implantation Failure

**DOI:** 10.3390/ijerph20053942

**Published:** 2023-02-23

**Authors:** Onder Celik, Arzu Yurci, Aynur Ersahin, Nur D. Gungor, Nilufer Celik, Mustafa D. Ozcil, Serdar Dogan, Semih Dalkilic, Lutfiye Dalkilic, Ulun Ulug, Sudenaz Celik, Andrea Tinelli

**Affiliations:** 1Private Clinic Obstetrics and Gynecology, 64000 Usak, Turkey; 2In Vitro Fertilization (IVF), Andrology and Genetics Center, Memorial Bahcelievler Hospital, 34180 Istanbul, Turkey; 3Department of Obstetrics and Gynecology, Bahcesehir University Goztepe Medicalpark Hospital, 34732 Istanbul, Turkey; 4Department of Biochemistry, Behcet Uz Children’s Hospital, 35210 Izmir, Turkey; 5Department of Obstetrics and Gynecology, Faculty of Medicine, Mustafa Kemal University, 31060 Hatay, Turkey; 6Department of Medical Biochemistry, Faculty of Medicine, Mustafa Kemal University, 31060 Hatay, Turkey; 7Molecular Biology and Genetics Program, Department of Biology, Faculty of Science, Firat University, 23000 Elazig, Turkey; 8Department of Obstetrics and Gynecology, School of Medicine, Halic University, 34060 Istanbul, Turkey; 9Medical Faculty, Sofia University “St. Kliment Ohridski”, 1407 Sofia, Bulgaria; 10Department of Obstetrics and Gynecology, CERICSAL (Centro di RIcerca Clinica SALentino), “Veris Delli Ponti Hospital”, 73020 Lecce, Italy

**Keywords:** implantation failure, endometrium, homeobox gene, mRNA, protein, fertility outcome

## Abstract

Background: Homeobox genes A10 (HOXA10) and A11 (HOXA11), members of the abdominal B gene family, are responsible for embryonic survival and implantation. This study was planned to investigate whether endometrial injury alters the expression of both transcripts in women with implantation failure. Methods: A total of 54 women with implantation failure were divided into two equal groups as experimental (scratching) and sham (no scratching). Participants in the scratching group were exposed to endometrial injury in the mid-luteal phase, and those in the sham group were exposed to endometrial flushing. The scratching group, but not the sham group, underwent prior endometrial sampling. A second endometrial sampling was performed on the scratching group in the mid-luteal phase of the following cycle. The mRNA and protein levels of the HOXA10 and 11 transcripts were determined in endometrial samples collected before and after injury/flushing. Participants in each group underwent IVF/ET in the cycle after the second endometrial sampling. Results: Endometrial injury caused a 60.1-fold (*p* < 0.01) increase in HOXA10 mRNA and a 9.0-fold increase in HOXA11 mRNA (*p* < 0.02). Injury resulted in a significant increase in both HOXA10 (*p* < 0.001) and HOXA11 protein expression (*p* < 0.003). There was no significant change in HOXA10 and 11 mRNA expressions after flushing. Clinical pregnancy, live birth, and miscarriage rates of the both groups were similar. Conclusions: Endometrial injury increases homeobox transcript expression at both mRNA and protein levels.

## 1. Introduction

Implantation success is defined by either a quantifiable increase in human chorionic gonadotropin (hCG) or the presence of sonographically demonstrable gestational sac. Implantation failure is defined as the absence of increased hCG levels or the failure of ultrasonographic visualization of the gestational sac despite increased hCG [1]. Although extensive advances in ovarian stimulation protocols and embryology laboratory conditions have led to a remarkable improvement in implantation rates, challenges still remain in recurrent implantation failure [2]. Maternal age, embryo quality, and endometrial pathologies are the basic parameters that should be investigated in cases with difficulty in treatment. The endometrium is an option that can be more easily intervened in according to age and embryo quality. However, since the data on how the endometrium performs its selectivity and receptivity functions are not sufficient, the search for new treatments continues [1,3]. Clinicians dealing with assisted reproductive techniques have focused their efforts on the endometrium to find a sustainable solution to implantation failure. In 2000, Granot and Barash collected endometrial samples on different days of the spontaneous menstrual cycle to investigate connexin 43 expression in patients with implantation failure [4]. The coincidental increase in pregnancy rates after the study led the authors to interpret that the endometrial damage that occurred during the endometrial sampling increased the implantation rates. In the study they planned in 2003 to confirm their hypothesis, the authors reported that endometrial injury nearly doubled the pregnancy rates in patients undergoing in vitro fertilization (IVF) [5].

After the pioneer studies of Barash et al. [4,5], studies conducted by other researchers differed in terms of their results. While some studies have reported that endometrial injury increases the rates of clinical pregnancy and live birth, others have reported that injury has no effect on fertility outcome [6,7,8]. The inconsistency of the results was attributed to the heterogeneity of the patient groups, the difference in the cycle phase in which the injury was performed, or the differences in the injury methods. Since endometrial sampling is a minimal invasive procedure, most studies have investigated the effect of injury on implantation and clinical pregnancy rates, rather than possible molecular changes in the endometrium. Putting forward hypothetical mechanisms instead of investigating the possible implantation promoting effect of injury has prevented the method from being used routinely. A limited number of studies have suggested that changes in the endometrium after injury are similar to the wound healing process in the menstrual cycle [9,10]. In connection with this, the authors suggested that increased production of growth factors, proinflammatory cytokines, and receptivity modulators during the post-injury recovery period increases implantation rates [4,5,9,11].

Up- or downregulation in the expression of different genes have been reported in endometrial microarray analyses performed after endometrial injury [12,13]. However, none of the genes whose expression has been reported to be altered is specific for receptivity. It is obvious that there is a need for studies that will explain the bio-molecular pathways behind the fertility-enhancing mechanism of endometrial injury by investigating its effect on receptivity modulators. The most important transcripts, whose defective expression is known to cause infertility in mice and humans, are homeobox genes [14]. These genes, which are members of the abdominal B (AbdB) gene family, are responsible for embryonic survival and implantation. The AbdB gene HOXA10 and HOXA11, whose expression increases during the peri-implantation period provides a successful implantation [14]. Mice with homozygous mutations in the HOXA10 or HOXA11 transcripts are mostly sterile despite healthy ovulation. In animals with targeted deletions for HOXA10 or HOXA11, embryos die early after implantation [14,15,16].

Mechanical endometrial injury has been put forward by many clinicians as an easy and inexpensive procedure capable of enhancing endometrial receptivity. Despite its widespread use in IVF practice, the effect of endometrial scratching on the expression pattern of receptivity genes is still an unresolved issue [9,11,13]. It is critical to determine the changes in the expression pattern of homeobox genes before and after the endometrial injury, in order to find a clear answer to the question of whether we should make a mechanical endometrial injury or not in women with implantation failure. Since studies showing that endometrial injury increases implantation rates are in the majority [6,7], it may increase the expression of receptivity modulators. We hypothesized that the improvement detected in fertility outcome following endometrial injury may be secondary to increased expression of receptivity genes. This study was therefore planned to investigate whether endometrial injury alters endometrial HOXA10 and HOXA11 expression in women with implantation failure. The mRNA and protein levels of these two transcripts were determined in endometrial samples collected before and after injury.

## 2. Materials and Methods

Study design and participants

Fifty-four 24- to 33-year-old sub-fertile women with recurrent implantation failure (RIF) who had menstrual cycles of 28 to 30 days and scheduled for local endometrial injury before ART were included in the study. All patients were noted to respond well to ovarian stimulation in their previous trial. This study was conducted in patients followed up with the diagnosis of implantation failure between July 2021 and May 2022 in the Kayseri Memorial Hospital IVF Center. Following the approval of the institutional ethics committee of MKU (Ethics Approval Number: 4298783/050-36/2021), participant recruitment started. 

There is no clear consensus on the definition of RIF. We considered the most commonly used definition “two or more failed cycles” as RIF [17]. Based on 0.05 alpha and 0.80 power, the required sample size for each group was determined as 30, using 0.05 ratios. Eligible women were fully counseled about endometrial scratching and their consent was obtained. Participants were non-randomly divided into two equal groups, experimental (scratching) and control (sham/non-scratching). Six participants from the injury group and twelve participants from the control group were excluded due to unacceptable RNA quality. Therefore, the number of participants, which was 27 at the beginning, decreased to 21 patients in the injury group and 15 patients in the control group when measuring mRNA (Figure 1). Demographic data, hormones, biochemical parameters, and fertility outcomes were calculated as 27 patients in each group. Patients over 40 years of age or with uterine or adnexal pathologies such as endometrioma, adenomyosis, endometrial polyp, or hydrosalpinx were excluded. Participants with hormone-secreting ovarian cysts, endometrial adhesions, PCOS, or submucosal or intramural fibroids with or without distorting the endometrium were excluded. Patients who had undergone endometrial scratching or curettage in their previous cycles were also excluded. For hormonal analysis, blood samples were collected from the patients in both groups on the 3rd day of the cycle before and after injury/flushing. Mid-luteal endometrial thickness was also recorded before and after injury/flushing.

Sample collection before and after endometrial injury/flushing

Scratching was performed in the mid-luteal phase of the menstrual cycle. Patients who had endometrial tissue sampling just prior to injury were subsequently exposed to endometrial injury. The scratching procedure was performed as previously described [5,8,18] using the pipelle catheter. The catheter was passed through the cervical canal and pushed until it touched the uterine fundus. The piston of the catheter was drawn back to the end of the sheath to create a negative pressure. The sheath was rotated 360 degrees and moved back and forth between the fundus and internal os at least 3–4 times before it was withdrawn to ensure endometrial injury has been created. The endometrial tissue samples collected from injury group were washed with a sterile saline solution to remove blood, transferred into RNA stabilization buffer (RNAlater; Qiagen, Hilden, Germany), and stored until analysis. If we had directly performed endometrial sampling from patients in the control group, the expression of receptivity genes might have changed due to local damage during sampling procedure. We, therefore, decided to make an endometrial flushing to collect endometrial cells from the patients in the sham group [18]. Endometrial flushing was applied to patients in the sham group in the mid-luteal phase of menstrual cycle. After cleaning the cervix with sterile saline solution, a catheter was passed through the cervix and the fundus was reached. Then, with the help of an injector, 5 mL of sterile saline was given to the endometrial cavity, and after a while, the given saline was aspirated back into tube containing RNAlater. Post-injury/flushing endometrial tissues were collected using a pipelle catheter during mid-luteal phase of the next menstrual cycle of patients in both groups.

The primary outcome was to analyze the mRNA and protein levels of homeobox genes before and after injury/flushing. The study group suffered two injuries (first intentional injury, second unintentional injury due to endometrial sampling) and the control group suffered one injury (unintentional injury due to endometrial sampling). We compared the changes in expression levels of homeobox transcripts before and after injury in the scratching group. In the sham group, we compared the expression levels of homeobox genes before and after flushing. The patients in the scratching group went to IVF/ICSI after the second endometrial sampling, and the patients in the flushing group in the cycle after the first endometrial sampling. Antagonist protocol was applied to the participants in both groups. On the fifth day, high quality single blastocyst transfer was performed in both groups. The secondary outcome was the comparison of clinical pregnancy, live birth, and miscarriage rates in both groups. Patients excluded due to insufficient RNA isolation were also included in the calculation of fertility outcomes because they were exposed to injury and underwent IVF/ET. A clinical pregnancy was defined by the presence of an intrauterine gestational sac confirmed by transvaginal ultrasound. In accordance with the definition of the World Health Organization, we defined a birth at or above 20 weeks of gestation as live birth [19]. The loss of fetus before 12 weeks of gestation was defined as a miscarriage. Clinical pregnancy, miscarriage, and live birth rates per cycle of fresh ET were calculated using the formulas below. Clinical pregnancy rate = total number of clinical pregnancies/total number of ET cycles × 100%; Live birth rate = total number of live births/total number of ET cycles × 100%; Miscarriage rate = total number of miscarriages/total number of clinical pregnancies × 100%.

RT-PCRSample preparation, RNA isolation, and cDNA synthesis

Endometrial samples in the RNAlater (Qiagen, Hilden, Germany) were homogenized with Tissuelyser (Qiagen, Hilden, Germany). Homogenized samples were used for total RNA isolation. PureLink RNA Mini Kit (Thermo Fisher Sci., Waltham, MA, USA) was used for isolation. PureLink DNase Set (Thermo Fisher Sci., Waltham, MA, USA) was used for removal of residual genomic DNA contamination. Both quantity and purity of RNA were detected with Qubit Fluorometer and Qubit HS RNA Assay Kit (Thermo Fisher Sci., Waltham, MA, USA). Quality of RNA samples were controlled with agarose gel electrophoresis. High-Capacity cDNA Reverse Transcription Kit (Applied Biosystems, Waltham, MA, USA) was used for obtaining complementary DNA (cDNA). The reaction mixture was incubated at 25 °C for 10 min and then was kept at 37 °C for 120 min and 85 °C for 5 min to inactivate RT.

Measurement of HOXA10 and HOXA11 mRNA expression

Primers of HOXA10 and HOXA11 genes were synthesized with the use of PerlPrimer. Sequences of all primers designed to be used as forward and reverse primers for RT-PCR were; HOXA-10: Forward 5′-GGT TTGTTC TGA CTT TTTGTT TCT-3′, Reverse 5′-TGA CAC TTA GGACAATAT CTATCTCTA-3′, HOXA-11: Forward 5′-AGT TCT TTCTTCAGC GTC TAC ATT-3′, Reverse 5′-TTT TTC CTT CAT TCTCCT GTTCTG-3′, GAPDH: Forward 5′-GAA GGT GAA GGT CGG AGT C-3′, Reverse 5′-GAA GAT GGT GAT GGG ATT TC-3′. Glyceraldehyde-3-Phosphate Dehydrogenase (GAPDH) was used as endogenous housekeeping gene. The mRNA levels of endometrial samples were normalized according to the GAPDH mRNA level. RT-PCR reaction was performed with the use of AMPLIFYME SYBR Universal Mix (BLIRT S.A., Gdańsk, Poland) and the Step One Plus (Applied Biosystems Waltham, MA, USA) real-time PCR device. The real-time PCR step was performed in 10 µL, including 5 µL of 2X concentrate Amplifyme SG Universal Mix, 0.3 µL forward primer, 0.3 µL reverse primer, 0.2µL 50X concentrate High ROX Solution, 2.2 µL PCR grade water, and 2 µL of cDNA sample. Thermal cycler was set to one cycle at 95 °C for 3 min, followed by 40 cycles of 95 °C for 5 s, 60 °C for 10 s, and 72 °C for 20 s. After, a melting curve analysis was performed for the accuracy of the PCR amplification.

Measurement of endometrial HOXA10 and HOXA11 protein expressions

All endometrial samples were diluted ½ before the assay. After the biopsy specimens were homogenized using a Tissuelyser (Qiagen) the flushing specimens were evaluated without homogenization. HOXA10 protein concentrations were assayed using commercially available kit by ELISA method (Thermoscientific Multiscan Go, Vantaa, Finland; Elabscience, Maryland, USA; catalog no: E-EL-H5498). The assay ranges for the HOXA10 kit were 15.63–1000 pg/mL, sensitivity 9.38 pg/mL, and the intra- and interassay coefficients of variance (CV%) were <10%. HOXA11 concentrations were assayed using an ELISA kit (Thermoscientific Multiscan Go Vantaa, Finland; Finetest, Wuhan, China; catalog no: EH-9185). The assay ranges for the HOXA11 kit were 15.62–1000 pg/mL, sensitivity 9.37 pg/mL, and the intra- and interassay coefficients of variance (CV%) were <10%. 

Statistical Analysis

RT-PCR results were expressed as Ct (cycle threshold), ΔCt, and ΔΔCt. Endometrial samples were studied three times and average Ct values were calculated according to relative quantification method. All Ct values exported as txt format and analyzed with DataAssist Software v3.01 (Applied Biosystem). The mRNA levels were calculated using the comparative ΔCt method (Ct of target gene–Ct of reference gene). The relative changes of mRNA expression levels of the HOXA10 and HOXA11 transcripts were calculated using the 2^−ΔΔCt^ method (ΔCT treated–ΔCT untreated), and obtained ΔCt and 2^−ΔΔCt^ data was used for statistical analysis. The Kolmogorov Smirnov test was used to evaluate the normality of the data distribution. Student’s *t* test was used for normally distributed variables, chi-square test for categorical variables, and Mann–Whitney test for non-normal variables. Continuous variables were presented as mean ± SD and categorical variables were presented as the number of cases or percentage. *p* < 0.05 was considered statistically significant.

## 3. Results

There was no significant difference between the two groups in terms of baseline characteristics shown in Table 1.

Pre-injury age, BMI, infertility duration, and the number of failed IVF/ICSI attempts of the two groups were similar. In addition to the serum FSH, LH, and estradiol levels detected on the 3rd day of the cycle, the AFCs of the patients in both groups were also similar. There was no significant difference between the serum FSH, LH, estradiol, and AFC determined before and after scratching/flushing. The pre-procedural endometrial thicknesses of the scratching and flushing groups were similar. Scratching caused a significant increase in endometrial thickness (*p* < 0.01). No significant change was detected in the endometrial thickness after flushing (*p* < 0.36). Mechanical injury was successfully performed in all participants in the scratching group, and endometrial flushing in the sham group. There were no significant complications related to the procedures performed in the scratching and sham groups. In a small number of patients with cramp-like pain after injury, the pain was relieved by expectant management or pain killers. Two participants in the scratching group experienced spotting the day following the procedure.

### 3.1. Pre- and Post-Injury Endometrial HOXA10 and 11 mRNA Levels

Since adequate RNA isolation could not be performed in 6 patients in the injury group and 12 patients in the control group, the number of participants whose mRNA was measured were 21 in the injury group and 15 in the control group (Figure 1). When compared with expression values before injury, a 60.1-fold increase was found in HOXA10 mRNA levels after injury (Table 2).

The fold-increase in HOXA10 mRNA after injury was statistically significant (*p* < 0.01). After endometrial injury, HOXA11 mRNA expression increased 9.01-fold compared to pre-injury values. This upregulation in HOXA11 mRNA was statistically significant (*p* < 0.02). After injury, there was an approximately six-fold increase in HOXA10 mRNA compared to HOXA11 (60.1-fold vs. 9.01-fold, *p* < 0.01). The heatmap plot shows the differences in expression levels of the HOXA10 and HOXA11 transcript for each sample (Figure 2).

Compared with the expression values before endometrial flushing in the non-scratching group, there was a 1.86-fold increase in HOXA10 mRNA expression after flushing. However, this upregulation in HOXA10 mRNA was not statistically significant (*p* < 0.40). A 0.3-fold decrease was detected in HOXA11 mRNA expression after flushing compared to expression values before flushing. This downregulation of HOXA11 mRNA levels was not statistically significant (*p* < 0.39). The initial (pre-injury) HOXA10 and HOXA11 mRNA levels of the both scratching and non-scratching groups were similar. The fold increase in HOXA10 mRNA after injury was significantly higher than the fold increase after flushing (60.1-fold vs. 1.86-fold, *p* < 0.01). While there was a 9.01-fold increase in HOXA11 mRNA levels after injury in the scratching group, there was a 0.30-fold decrease in HOXA11 mRNA expression after flushing in the non-scratching group. This difference in HOXA11 mRNA values between the two groups was statistically significant (9.01-fold vs. 0.30-fold, *p* < 0.03; Table 2 and Figure 3).

### 3.2. Pre- and Post-Injury Endometrial HOXA10 and HOXA11 Protein Levels

When compared with the values before injury, there was a nearly three-fold and significant increase in endometrial HOXA10 protein levels after injury (585.3 ± 241.6 pg/mL vs. 1551.9 ± 304.8 pg/mL, *p* < 0.001). There was a significant increase in HOXA11 protein levels after injury compared to pre-injury values (969.3 ± 226.3 pg/mL vs. 1256.6 ± 121.1 pg/mL, *p* < 0.003). In the non-scratching group, there was no significant change in endometrial HOXA10 and 11 protein levels during and after flushing (Figure 3). The changes detected in HOXA10 and 11 protein levels before and after the injury/flushing in both the scratching and non-scratching groups were consistent with the changes detected in mRNA (Table 3).

### 3.3. Fertility Outcome following Injury/Flushing

No significant difference was detected in the cycle characteristics of both groups (Table 1). Clinical pregnancy (12/27 (44.4%) vs. 10/27 (37.0%) *p* = 0.544) and live birth rates of the both groups were similar (9/27 (33.3%) vs. 8/27 (29.6%) *p* = 0.09). A non-significant increase in clinical pregnancy and live birth was observed in the scratching group. Similarly, both groups had similar miscarriage rates (3/12 (25%) and 2/10 (20%), *p* = 0.40).

## 4. Discussion

The inability to clearly demonstrate the biological mechanisms that increase implantation and pregnancy rates is the most important factor limiting the use of endometrial injury. While mechanical endometrial injury caused a significant increase in implantation, clinical pregnancy and live birth rates in patients with recurrent implantation failure [6,20,21,22], the same positive effect could not be demonstrated in the unselected patient groups [8,23]. Considering the moderate quality of the studies conducted so far, and differences in injury methods and phases of injury, there is not enough convincing data that it is beneficial to make endometrial injury in any patient group, including RIF [24,25,26]. To determine the expression change in endometrial receptivity genes (HOXA10 and HOXA11) after injury we compared the endometrium of injury-treated and -untreated women with a history of failed implantation. We showed for the first time that mechanical injury caused a 60-fold increase in HOXA10 mRNA and a 9-fold increase in HOXA11 mRNA. Both HOXA10 and HOXA11 protein levels increased significantly after injury. The simultaneous increase in homeobox mRNA and protein expression is evidence that injury improves implantation through these transcripts. The tendency of clinical pregnancy and live birth rates to increase in the scratching group compared to the control group supports this idea. The similar fertility outcomes of the twice-injured group and the single-injured group suggest that the number of injuries is not very critical in determining the fertility outcome. In fact, the trend of increasing miscarriage rates in the group that was injured twice suggests that repetitive scratching may be harmful. However, involuntary injury occurring during endometrial sampling did not exhibit the stimulatory effect of conventional injury on homeobox genes. In the control group with unintentional injury, we did not detect significant changes in HOXA10 and HOXA11 mRNA and protein levels before and after flushing. Endometrial flushing caused 1.86-fold upregulation in HOXA10 mRNA and 0.30-fold downregulation in HOXA 11 mRNA. However, the upstream and downstream changes in both transcripts were insignificant. Although there was no significant increase in receptivity gene expression in the control group, having a fertility outcome similar to the injury group suggests that receptivity modulators other than receptivity genes also positively affect pregnancy rates [11].

To date, no studies have been conducted to investigate the changes in the expression of receptivity genes before and after the injury. The first studies on the mechanism of action of mechanical injury were designed considering the similarity of the menstrual cycle and the post-injury healing process [5,9,11]. In fact, there are close similarities between post-injury healing and physiological post-menstrual endometrial remodeling. The healing process that occurs in the endometrium after menstruation occurs via cytokines and growth factors, whose production and release are regulated by ovarian steroid hormones. A similar healing process takes place after mechanical endometrial injury, but unlike normal menstruation, massive amounts of cytokines, growth factors, and inflammatory molecules are secreted [5,9,11]. On the other hand, since mechanical injury cannot alter ovarian sex steroid synthesis, changes in the damaged endometrium are largely independent of the effects of estrogen and progesterone. The fact that the increase in pregnancy rates after endometrial injury is limited to the next cycle suggests that the effect of injury is temporary and related to the local healing process [6,7,9]. In this study, an increasing trend was detected in fertility outcomes, although patients went to IVF/ICSI after the second endometrial sampling. While the effectiveness of injury is limited to the next cycle, unintentional injury due to the second endometrial sampling may have caused the injury-related changes to continue in the next cycle. Therefore, the improvement in pregnancy rates may be due to increased expression of receptivity genes or pro-inflammatory cytokines secondary to damage during the second endometrial sampling. In support of this, the improvement in decidualization after mechanical injury in animal models has been accepted as evidence of local effects of injury [27,28]. The fact that anti-inflammatory drugs inhibit decidual development in the rabbit model is another indication that post-injury healing is an inflammatory process [29]. On the other hand, increased synthesis of proinflammatory cytokines or growth factors alone is not sufficient for successful implantation. Simultaneously, there should be an increase in the expression of receptivity modulators. This study showed that local injury leads to a significant increase in the expression of homeobox genes, closing a large gap in the implantation puzzle. In addition, we also showed that performing the injury in the mid-luteal phase has a critical importance for the increase of receptivity gene expression. Our rates of clinical pregnancy, miscarriage, and live birth were similar to the Cochrane Library results published in 2021 [30].

Homeobox genes provide the coordination between receptivity modulators required for implantation. HOXA10 and 11 begin to be expressed in the normal menstrual cycle, first in the luminal and glandular epithelium of the endometrium, and then in stromal cells [9,14,15]. This gradual increase in homeobox transcript expression is essential for healthy endometrial growth, differentiation, and decidualization [31,32]. Furthermore, both genes also regulate the release of growth factors and cytokines necessary for implantation and embryo survival [15,16]. About 80% of female mice with homozygous HOXA10 mutations are sterile. In these animals, between 2.5 and 3.5 days post-coital, embryos die before they can be implanted [15,16]. For these reasons, HOXA10 mRNA, of which we detected 60-fold increase in expression and HOXA11 mRNA, which increased 9-fold, may be one of the main mechanisms underlying the increase in pregnancy rates in the following cycle. The increase in mRNA expression accompanied by the increase in protein expression is evidence that the injury stimulates both transcriptional and translational events. Massive increase in HOXA10 mRNA and moderate increase in HOXA11 mRNA after injury may enhance embryo survival by coordinating the increase in cytokines, growth factors and pro-inflammatory cytokines. Consistent with this idea, Gnainsky et al. [11] showed that injury induces the production of inflammatory cytokines such as tumor necrosis factor-α, interleukin-15, and macrophage inflammatory protein 1B. Although a physiological amount of inflammation is necessary for successful implantation, the quality of the transferred blastocyst is another important determinant of implantation. Since we transferred a single and high-quality blastocyst to both groups, we can attribute the improvement in fertility outcome to the increase in receptivity genes and proinflammatory cytokine expression. Failed decidual development of Hox mutant female mice despite adequate estrogen and progesterone supports the critical role of homeobox genes in ensuring coordination between endometrial cytokines [12,13,14].

We did not detect any significant changes in serum estradiol and AFC after injury. This finding is important evidence that the changes that occur in the endometrium in the post-injury period are local and do not cause a change in the ovarian reserve and the production of sex steroids. HOXA10 and HOXA11, whose expression is increased in the post-injury period, may be rearranging the effects of estrogen and progesterone on the endometrium. This idea is supported by the fact that HOXA10 coordinates the response of endometrial stromal cells to progesterone during decidualization and implantation [33]. Consistent with this, it has been reported that advanced endometrial maturation due to supra-physiological progesterone increase in stimulated cycles improves with local injury [34,35,36]. In addition, since endometrial damage is known to improve estrogen receptor expression, we might suggest that homeobox genes mediate improvement in implantation rates via steroid hormone receptors [34,35]. Studies investigating the effect of injury on the endometrium in animal models whose ovaries were surgically removed are critical in terms of revealing how the implantation promoting effects of the mechanical injury occur in the absence of estrogen and progesterone.

The endometrium of the patients in both groups was of sufficient thickness by ultrasonography. However, the fact that the endometrium looks good on ultrasonography and is of sufficient thickness does not mean that it meets all the conditions necessary for implantation [37]. The fact that homeobox gene expression is impaired in patients with endometrioma, PCOS or hydrosalpinx despite normal-appearing endometrioma supports this idea [37]. For example, defective endometrial HOXA10 mRNA expression of women with hydrosalpinx has been shown to return to normal after salpingectomy [38]. In a recent study conducted by our team, a significant increase was found in HOXA10 and 11 mRNA expressions after endometrioma resection [31]. We also showed that failed endometrial HOXA10 and 11 mRNA expressions were normalized after laparoscopic ovarian drilling in PCOS [39]. There is no study on whether homeobox gene expressions are changed in patients with recurrent implantation failure. A recent study reported an 11-fold increase in endometrial leukemia inhibitory factor (LIF) mRNA expression after mechanical injury [8]. It is known that the defective expression of LIF disrupts the release of type 2 T-helper cytokines and causes recurrent abortions [40,41]. Although LIF expression is normal in mice with HOXA10 mutant uterus, embryo death and resorption in the early post-implantation period [16] suggest that HOXA10 is required for successful implantation [16,42]. The fact that the miscarriage rates were similar between the injury and the control group suggests that local injury does not have a significant effect on early fetal loss. However, the limitation in the number of participants prevents us from making a clear interpretation of the relationship between scratching and miscarriage.

Although it is known that mechanical injury changes the expression of both cytokines and genes in the endometrium [11,13], we do not know by which mechanism injury increases the expression of homeobox genes. Both HOXA10 and 11 expressions may be increased due to the traumatic effect of the catheter or concurrently with the increase in cytokines during the wound healing process [11,12,13]. The fact that homeobox genes increase the production of cytokines that will ensure embryo viability is evidence supporting our idea of simultaneous release [15]. Consistent with the last statement, in endometrial specimens examined by microarray after local injury, one study [12] reported increased expression in 218 genes and another study [13] in 183 genes. Authors suggested that local injury may increase implantation rates, particularly by modulating the expression of endometrial uroplakinIb, MUC1, laminin alpha 4, integrin alpha 6 and matrix metalloproteinase 1 [12,13]. However, none of the genes shown in microarray studies were specific receptivity genes. A 60-fold increase in HOXA10 mRNA and a 9-fold increase in HOXA11 mRNA after injury is evidence that homeobox genes contribute to the restoration of damaged endometrial tissue. In the mechanical injury group, the pre-injury HOXA10 and 11 transcripts reached the threshold value (Ct) at the 33^rd^ thermal cycle, while the number of thermal cycles required reaching the threshold after the injury was 26. This finding suggests that there is a rapid increase in HOXA10 and 11 expressions after injury compared to pre-injury values. This increase may be due to surgical stress during the injury or to uterine manipulations during the procedure. It is known that surgical interventions on the ovaries or fallopian tubes [43] cause changes in the expression of some genes in the endometrium. However, the 1.86-fold insignificant upregulation in HOXA10 mRNA and 0.30-fold downregulation in HOXA11 mRNA after endometrial flushing and the fact that HOXA10 and 11 reached the threshold value (Ct) late in the thermal cycle suggests that uterine manipulations, catheter placement (mock cycle), or injecting pressurized saline did not cause a significant increase in homeobox gene expressions. Therefore, we can attribute the increases in HOXA10 and HOXA11 mRNA to the mechanical injury procedure itself. Contrary to classical injury, the lack of increase in receptivity gene expression in the flushing group strongly suggests that simple uterine manipulations such as mock embryo transfer or endometrial flushing do not have a fertility improvement effect. The fact that mock embryo transfer performed before embryo transfer or during oocyte retrieval has been reported to have no significant effect on the fertility outcome supports this idea [44,45].

## 5. Conclusions

Since the increase in pregnancy rates after intentional endometrial injury is not supported by biological mechanisms, there is no clear recommendation for the routine use of the scratching. Despite the small number of participants and all our shortcomings in the study design, we showed for the first time that endometrial injury led to a significant increase in both mRNA and protein expression of homeobox genes. In the light of previous studies and our results, we can suggest that the expressions of receptivity modulators in women with different infertility etiologies are defective and return to normal with the treatment of the underlying pathology [31,37,38,39]. Although the maximum increase in the homeobox gene was reported 17-fold [31] after endometrioma resection, we found a 60-fold increase following mechanical injury. This finding suggests that direct interventions into the endometrium stimulate receptivity gene expression more effectively. The remarkable increase in HOXA10 and 11 mRNA and protein after mechanical injury suggests that the cells responsible for the synthesis and release of receptivity genes in women with implantation failure are impaired and become functional again with mechanical injury. We believe that our results will fill an important gap in the mechanism of action of endometrial injury. On the other hand, due to the minimal number of participants in both the scratching and control groups, this study is not powerful enough to draw a robust conclusion about the routine use of this procedure in women with implantation failure. It is hoped that future studies will overcome the limiting factor of the small number of participants and include injury-related biochemical changes, but this study provides a very strong foundation.

## Figures and Tables

**Figure 1 ijerph-20-03942-f001:**
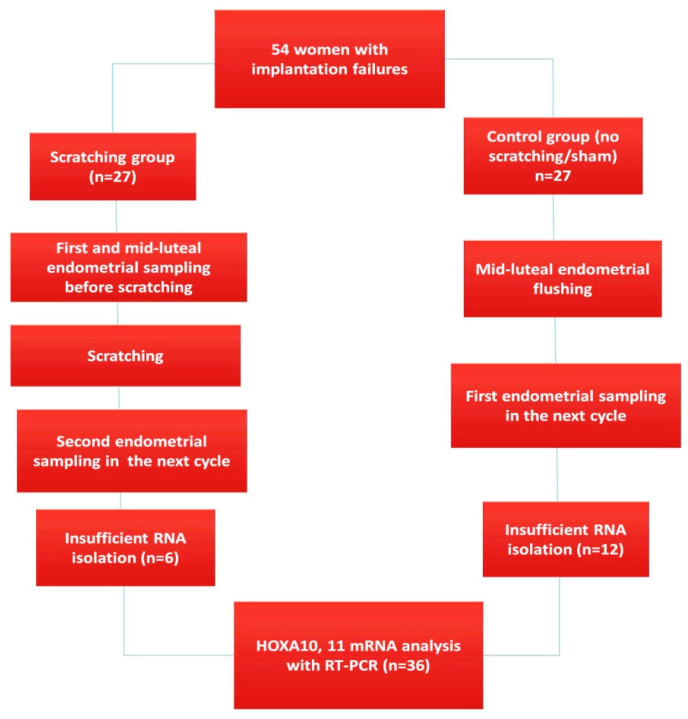
Patient selection flowchart representation of the participants in scratching and non-scratching groups.

**Figure 2 ijerph-20-03942-f002:**
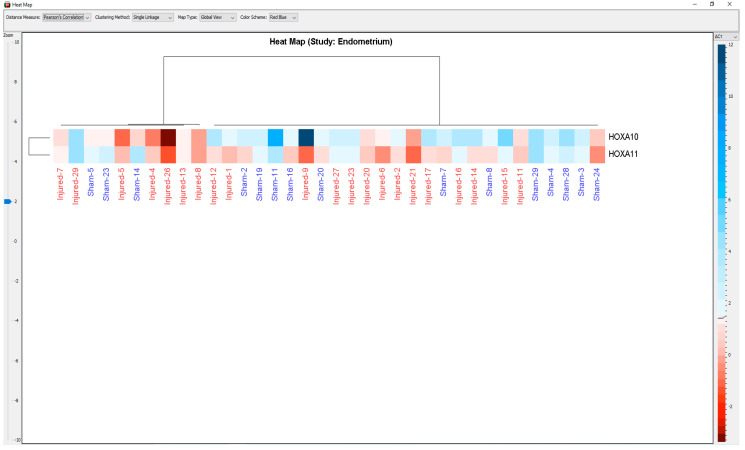
The heatmap plot shows the differences in expression levels of the HOXA10 and HOXA11 genes for each sample. Distance was measured according to Pearson Correlation and Single Linkage clustering method was applied during analysis. All *p* values were adjusted according to the Benjamin–Hochberg False Discovery Rate.

**Figure 3 ijerph-20-03942-f003:**
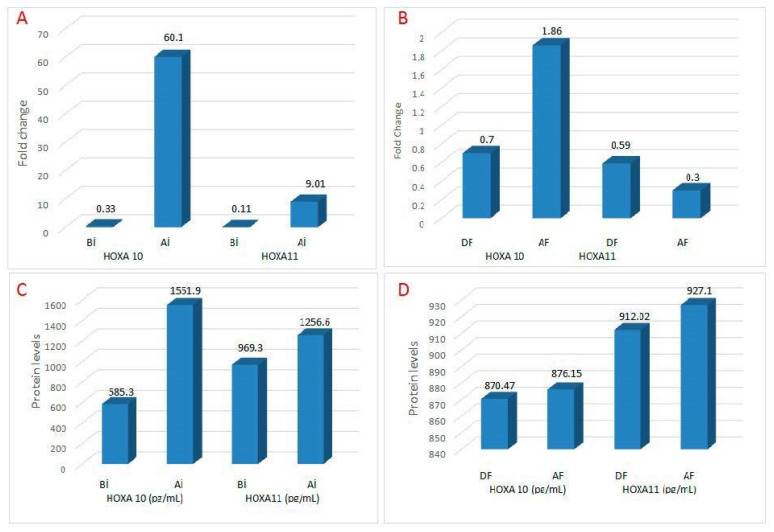
Relative quantification (fold change) graph show expression differences of HOXA10 and HOXA11 transcripts between scratching and sham (non-scratching) groups. (**A**) In the scratching group, the injury caused a 60-fold increase in HOXA10 mRNA and a 9-fold increase in HOXA11 mRNA. (**B**) In the sham group, endometrial flushing did not cause a significant change in HOXA 10 and 11 mRNA expressions. The changes detected in HOXA10 and 11 protein levels before and after the injury/flushing in both the scratching (**C**) and sham groups (**D**) were consistent with the changes detected in mRNA (BI means before injury, AI means after injury, DF means during flushing, and AF means after flushing).

**Table 1 ijerph-20-03942-t001:** Comparison of demographic characteristics of the scratching and sham group.

	Endometrial Scratching (n = 27)		Non-Scratching (n = 27)		*p* *	
Age (yrs)	28.3 ± 5.22		27.9 ± 4.20		0.26	
BMI (kg/m^2^)	22.9 ± 1.53		23.7 ± 4.13		0.32	
Infertility duration (yrs)	3.76 ± 1.40		3.89 ± 1.02		0.09	
Failed IVF/ICSI trial	2.79 ± 1.50		2.87 ± 1.90		0.43	
	Before scratching	After scratching	*p*	During flushing	After flushing	*p*
Endometrial thickness (mm)	8.66 ± 2.76	10.9 ± 2.30	0.01	8.59 ± 1.99	9.02 ± 2.60	0.36
LH (mIU/mL)	4.97 ± 1.90	5.13 ± 1.07	0.69	4.87 ± 1.20	4.98 ± 1.87	0.20
FSH (mIU/mL)	5.43 ± 1.08	5.32 ± 1.23	0.51	5.39 ± 1.10	5.40 ± 1541	0.64
Estradiol (pg/mL)	37.9 ± 6.40	39.4 ± 4.80	0.74	39.7 ± 6.70	38.3 ± 7.44	0.51
AFC	10.2 ± 4.22	10.9 ± 3.21	0.36	9.20 ± 3.40	8.76 ± 2.40	0.56

*: Data were presented as mean ± SD and *p* < 0.05 was considered statistically significant.

**Table 2 ijerph-20-03942-t002:** Comparison of average ΔCt, 2^−ΔCt^, and fold changes of HOXA10 and HOXA11 mRNA before and after endometrial injury.

Genes	Groups	Average ΔCt	Average 2^−ΔCt^	Fold Change *	*p*-Values **	Regulation
Scratching group (n = 21) ***						
1- HOXA10	a-Before injury	7.66	0.0010	0.33	0.51	Down
	b-After injury	6.03	0.0350	60.1	0.01	Up
2- HOXA11	c-Before injury	5.70	0.0044	0.11	0.30	Down
	d-After injury	4.30	0.0600	9.01	0.02	Up
Non-scratching group (n = 15) ***						
3- HOXA10	a-During flushing	8.50	0.0303	0.70	0.22	Down
	b-After flushing	7.01	0.0245	1.86	0.40	Up
4- HOXA11	c-During flushing	4.05	0.0003	0.59	0.24	Down
	d-After flushing	4.20	0.0350	0.30	0.39	Down
1a vs. 3a				0.33 vs. 0.70	0.08	
1b vs. 3b				60.1 vs. 1.86	0.01	
2c vs. 4c				0.11 vs. 0.59	0.65	
2d vs. 4d				9.01 vs. 0.30	0.03	
1b vs. 2d				60.1 vs. 9.01	0.01	

* Fold change ≥3 was accepted as positive (up) regulation for the gene studied, ** *p* < 0.05, *** Due to insufficient RNA isolation in the injury (n = 6) and sham (n = 12) groups, the number of participants in the groups changed to 21 and 15, respectively.

**Table 3 ijerph-20-03942-t003:** Concentrations of endometrial HOXA10 and HOXA11 proteins in the scratching and sham groups.

Homeobox Genes	Groups	Protein Levels *	*p*-Values **
Scratching (n = 27)			
HOXA10 (pg/mL)	Before injury	585.3 ± 241.6	<0.001
	After injury	1551.9 ± 304.8	
HOXA11 (pg/mL)	Before injury	969.3 ± 226.3	<0.003
	After injury	1256.6 ± 121.1	
Non-scratching (n = 27)			
HOXA10 (pg/mL)	During flushing	870.4 ± 221.2	>0.05
	After flushing	876.1 ± 231.0	
HOXA11 (pg/mL)	During flushing	912.0 ± 149.6	>0.05
	After flushing	927.1 ± 51.30	

* The protein levels are shown as mean ± SD, ** *p* < 0.05.

## Data Availability

If all the authors make a joint decision, data will be made available to the editor of the journal for review or query upon request. The data are not publicly available due to restrictions set by the human research ethics committee.

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
