# Peer review of "Endometrial Injury Upregulates Expression of Receptivity Genes in Women with Implantation Failure"

_ijerph, 2023, doi:10.3390/ijerph20053942_

Round 1

Reviewer 1 Report

"Endometrial injury up regulate expression of receptivity genes and improves fertility outcome in women with implantation failure" This paper is one among several studies tending to show a benefit effect of endometrial scratching before embryo transfer. This article is well writing and clear. This non randomized work , based on HOXA10 and 11 endometrial expression, like other past studies, has limited power to detect clinically relevant differences, with a lot of bias. Endometrial HOXA 10 and 11 are expressed also in endometriosis and multiple inflammatory situations. In the last Cochrane Library published in 2021, the effect of endometrial injury in live birth is unclear as the result is consistent with no effect( 4402 participants randomly studied). Endometrial injury results little to no difference to the rate miscarriage (OR 1.03, 95% CI 0.85 to 1.25; participants = 8092;studies = 30; I2 = 0%, low-certainty evidence). If the chance of miscarriage from IVF is normally 5.4%, then the chance of miscarriage from endometrial injury before IVF is between 4.6% and 6,7%

The authors have to explain the mechanism by which an endometrial injury one or two cycles before embryo transfer may change the cytokine profile of an endometrium under ovarian stimulation. The authors have also to mention that a simple mock transfer modify the cytokine profile of the endometrium with an increase of TNF alpha, IL6, HOXA 10 and others..) Despite small studies suggesting possible benefit from endometrial injury, due to risk of bias and heterogeneity between studies, there is no sufficient evidence to recommend routine use of this procedure in practice.

The authors have to consider the role of the blastocyst itself and the inflammatory process participating to the implantation. The blastocyst quality introduces another bias in the discussion.

Author Response

Q: In the last Cochrane Library published in 2021, the effect of endometrial injury in live birth is unclear as the result is consistent with no effect. Endometrial injury results little to no difference to the rate miscarriage.

R: Our rates of clinical pregnancy, miscarriage and live birth were similar to the Cochrane Library results published in 2021.

Q: The authors have to explain the mechanism by which an endometrial injury one or two cycles before embryo transfer may change the cytokine profile of an endometrium under ovarian stimulation.

R: The fact that the increase in pregnancy rates after endometrial injury is limited to the next cycle suggests that the effect of injury is temporary and related to the local healing process [5,6,8]. In this study, an increasing trend was detected in fertility outcomes, although patients went to IVF/ICSI after waiting for one cycle for the second endometrial sampling. While the effectiveness of injury is limited to the next cycle, endometrial damage secondary to the second endometrial sampling may have caused the injury-related changes to continue in the next cycle. The improvement in pregnancy rates may be due to the increase in receptivity gene or inflammatory cytokines expression secondary to injury during second endometrial sampling with a pipelle. Ovarian stimulation does not cause an adverse effect on endometrial healing after injury. In good agreement with, it has been reported that advanced endometrial maturation due to supraphysiological progesterone increase in stimulated cycles improves with local injury [30-32].

Q: The authors have also to mention that a simple mock transfer modifies the cytokine profile of the endometrium with an increase of TNF alpha, IL6, HOXA 10 and others).

R: Contrary to classical injury, it is known that mock embryo transfer does not affect IVF, implantation or pregnancy rates. Mock embryo transfer before embryo transfer or during oocyte retrieval does not exert a significant effect on the endometrium (kataria). It has been reported that conventional injury induces the production of inflammatory cytokines such as tumor necrosis factor-α, interleukin-15, and macrophage inflammatory protein 1B [10]. In this context, it can be thought that cytokine release due to endometrial contact will increase during the insertion of a catheter into the cavity in mock cycles. However, in order to be called an injury, almost the entire cavity must be subjected to a pressurized injury in the form of repetitive forward and backward pulling with the help of a catheter. In agreement with this we did not find any study on endometrial proinflammatory cytokine change in mock cycles. For receptivity gene changes in mock cycles, we can give an example from our study. The insignificant up-regulation in HOXA10 mRNA and down-regulation in HOXA11 mRNA in the samples taken after endometrial flushing in sham group without injury and the fact that HOXA10 and 11 reached the threshold value (Ct) late in the thermal cycle suggests that uterine manipulations, catheter placement (mock cycle) or injecting pressurized saline did not cause a significant increase in HOXA10 and 11 expressions.

Q: “Despite small studies suggesting possible benefit from endometrial injury, due to risk of bias and heterogeneity between studies, there is no sufficient evidence to recommend routine use of this procedure in practice”.

R: The last sentence of the article has been changed as follows in line with your suggestion. “Due to the minimal number of participants in both the scratching and control groups, this study is not powerful enough to draw a robust conclusion about the routine use of this procedure in women with implantation failure”.

Q: The authors have to consider the role of the blastocyst itself and the inflammatory process participating to the implantation. The blastocyst quality introduces another bias in the discussion.

R: Although a physiological amount of inflammation is necessary for successful implantation, the quality of the transferred blastocyst is another important determinant of implantation. Since we transferred good quality blastocysts to both groups, we can attribute the improvement in clinical pregnancy and live birth rates to the increase in receptivity genes and proinflammatory cytokines. The fact that Hox mutant female mice exhibit impaired decidualization despite adequate estrogen and progesterone supports the critical role of homeobox genes in ensuring coordination between endometrial cytokines during the implantation [11-13].

Reviewer 2 Report

The study is furthering the author's interest in examining the HOXA 10 and 11 mRNA and protein levels determined by surgery and mechanical intervention. It is their merit in designing the study so that it produces clear results that are statistically relevant. This work is paramount for improving fertility rates in patients with endometrial factors. Further studies hopefully will surpass the limiting factor of the small number of cases and will be closer homing on the biochemical interactions but this study lays a very strong foundation.

There are a few language corrections necessary> line 52 - unclear wording, line 81 'has' to be replaced with 'have

Author Response

Q: There are a few language corrections necessary> line 52 - unclear wording, line 81 'has' to be replaced with 'have

R: The entire article was edited by a native speaker for grammatical and typographical errors.

Reviewer 3 Report

This pilot study showed for the first time that mechanical endometrial injury led to a significant increase in both mRNA and protein expression of homeobox genes. This increase in expression of homeobox genes may be responsible for the increase in clinical pregnancy and live birth rates after injury in patients with implantation failure. 

The study is small, but well performed. My only objection is the low quality of the layout of the figures, which are blurred and unreadable. 

Author Response

Q: The study is small, but well performed. My only objection is the low quality of the layout of the figures, which are blurred and unreadable. 

R: The image quality of Figures 1 and 3 has been improved (If requested, we can send the originals of the figures).